# Implementation of *Mycoplasma genitalium* Diagnostics with Macrolide-Resistance Detection Improves Patient Treatment Outcomes in Bulgaria

**DOI:** 10.3390/diagnostics14232665

**Published:** 2024-11-26

**Authors:** Ivva Philipova, Maria Mademova, Elena Birindjieva, Venelina Milanova, Viktoriya Levterova

**Affiliations:** 1National Center of Infectious and Parasitic Diseases (NCIPD), 1504 Sofia, Bulgaria; 2Faculty of Biology, Sofia University St. Kliment Ohridski, 1164 Sofia, Bulgaria; 3Center for Sexual Health CheckPoint Sofia, 1000 Sofia, Bulgaria

**Keywords:** *Mycoplasma genitalium*, diagnostics, macrolide-resistance assay, resistance-guided therapy, Bulgaria

## Abstract

**Background/Objectives:** The increasing prevalence of *Mycoplasma genitalium* infections with macrolide-resistance, causing high azithromycin failure rates, is a major concern internationally. In response to this challenge, diagnostics that simultaneously detect *M. genitalium* and genetic markers for macrolide-resistance enable the therapy to be individually tailored, i.e., to implement resistance-guided therapy (RGT). This study aimed to evaluate patient treatment outcomes of *M. genitalium* therapy, guided by a macrolide-resistance assay in Bulgaria. **Methods:** Consecutively referred *M. genitalium* infection cases (*n* = 17) were analyzed for macrolide-resistance mutations (MRMs) and specific antimicrobial treatment was recommended accordingly (MRMs-negative infections received azithromycin and MRMs-positive infections received moxifloxacin). The treatment outcome based on test-of-cure was recorded, and the treatment failure rates and time to achieve a microbiological cure were compared to treatment outcomes in patients treated before the implementation of RGT. **Results**: Among patients given RGT (*n* = 17), the overall treatment failure rate was 1/17 (5.9%). This was significantly lower than the rate (47.6%) observed in patients treated pre-RGT (*p* = 0.002). The time to achieve a microbiological cure was 29.4 days (CI 24.5–34.3), compared to 45.2 days (CI 36.5–53.7) pre-RGT (*p* = 0.001). **Conclusions:** The implementation of *M. genitalium* diagnostics with macrolide-resistance detection improved treatment outcomes in Bulgaria, with significantly lower treatment failure rates and reduced time to achieve a microbiological cure. In light of the limited treatment options and concerns about their decreasing efficacy in response to misuse and overuse, a diagnostic macrolide-resistance assay is critical to direct the appropriate first-line treatment, to maintain the efficacy of antimicrobial treatment (antibiotic stewardship) and to minimize the spread of antimicrobial resistance.

## 1. Introduction

*Mycoplasma genitalium* causes non-gonococcal urethritis in men and urethritis, cervicitis and pelvic inflammatory disease in women [1,2,3,4,5]. If *M. genitalium* is left untreated, men have a very high risk of persistent or recurring urethritis symptoms and ascending infection may occur (i.e., epididymitis) [5]. In women, studies demonstrated an association between *M. genitalium* infection and tubal factor infertility [2,3]. Furthermore, infections during pregnancy can be associated with adverse pregnancy outcomes, such as spontaneous abortion and preterm birth [4]. The appropriate diagnostic methods of *M. genitalium* are limited to nucleic acid amplification tests (NAATs), as culture is extremely slow (several months), challenging and insensitive [6]. In the routine practice, the use of NAATs leads to the initiation of antimicrobial therapy without antimicrobial susceptibility testing, since no viable isolates are available for the subsequent testing [7]. Due to an inherent mycoplasmas resistance to many antimicrobial classes, treatment options are scarce and the European *M. genitalium* guidelines recommend azithromycin (0.5 g day 1, followed by 0.25 g days 2–5) as first-line treatment, while second-line treatment is moxifloxacin (400 mg per day, 7 days) [5]. For comparison, an increased azithromycin dose (1 g day 1, 0.5 g days 2–4) is used as the first-line treatment in Bulgaria, while the second-line moxifloxacin treatment is identical [8]. Current partners of *M. genitalium*-positive patients should be tested and treated with the same antimicrobial as the index patient [5]. Treatment is becoming even more challenging due to increasing antimicrobial resistance, especially to macrolides. Multiple studies report a high prevalence of resistance and treatment failures across the globe [5,9,10,11]. For instance, in most European countries, North America, East Asia and Australia, the increasing treatment failures with azithromycin in *M. genitalium* over the last 15 years have been consistent with the increases in the prevalence of macrolide-resistance mutations from 10% before 2010 to more than 50% in 2024 [10,12]. Although the presence of quinolone-resistance mutations leads to treatment failure in only around 60% of the infected patients, in the last five years these mutations have increased in prevalence worldwide, from 8% in 2019 to more than 20% in some countries in Southern and Southeast Europe, and even more than 80% in East Asia in 2024 [10,11,12]. Therefore, *M. genitalium* was recognized as an emerging global public health threat by the US Centers for Disease Control and Prevention [13]. Addressing this threat requires regular updates on the extent of antimicrobial resistance and the slowing of its spread through optimized approaches for diagnostics and treatment (antimicrobial stewardship) until new antibiotics are developed [14]. An innovative approach in the diagnostics of *M. genitalium* involves simultaneous detection of both the pathogen and mutations associated with antimicrobial resistance. The additional information about resistance status allows for the utilization of resistance-guided therapy (RGT) for prescribing an antimicrobial that is most likely to treat the particular strain of infection. The RGT of *M. genitalium* has been implemented in Australia, the United Kingdom and Germany, and has demonstrated improvements in the cure rate, treatment time and cost [15,16,17,18]. Furthermore, it is a valuable tool helping to overcome the global threat of antibiotic resistance. In Bulgaria, a high azithromycin failure rate (47.6%) has been observed and patients experience a lengthy time to achieve a cure, including multiple clinic visits and antibiotic courses [19]. An intervention to improve patient treatment outcomes in the context of the country’s widespread resistance and high first-line treatment failure rates thus became a necessity. Therefore, to guide first-line treatment, *M. genitalium* diagnostics with macrolide-resistance detection was implemented at the beginning of 2022.

This study aimed to evaluate patient treatment outcomes of *M. genitalium* therapy guided by a macrolide-resistance assay in Bulgaria by comparing (1) treatment failure rates and (2) the mean time to achieve a microbiological cure, before and after its implementation.

## 2. Materials and Methods

### 2.1. Study Design and Population

This was a prospective case study analysis of patients given macrolide RGT for *M. genitalium* infection between 1 January 2022 and 31 December 2022 at the National Center of Infectious and Parasitic Diseases in Sofia, Bulgaria. Testing was performed on referred attendees of the sexual health clinic “CheckPointSofia”, for whom *M. genitalium* testing was indicated as follows: presentation with symptoms and signs and/or risk factors for infection. The enrolled participants were patients who were diagnosed with an *M. genitalium* infection in 2022 and received specific antimicrobial treatment according to pathogen resistance status (resistance-guided therapy). Specifically, patients with *M. genitalium* infection in the absence of macrolide-resistance mutations (MRMs) received azithromycin (1 g day 1, 0.5 g days 2–4) and those with infection with the presence of MRMs were treated with moxifloxacin (400 mg per day, 7 days). Participants were asked to return for a test-of-cure (TOC) 21 days after completing the recommended antimicrobial therapy and to refrain from sexual activity until a negative result was obtained. At the TOC, patients were assessed for therapy compliance and reinfection risk, and cases with non-adherence to the recommended dosing regimen and suspected reinfection were excluded. In case of treatment failure, patients were treated according to the recommendations of the European *M. genitalium* guidelines, i.e., moxifloxacin (400 mg per day, 7 days) for persistent *M. genitalium* infection in the presence of selected macrolide-resistance after azithromycin treatment, and pristinamycin (1 g four times daily for 10 days) for persistent infection after moxifloxacin treatment [5].

### 2.2. Laboratory Testing

The routine diagnostics (standard care) were performed with Real-Time PCR (Mycoplasma genitalium Real-TM assay, Sacace Biotechnologies s.r.l., Como, Italy), as per manufacturer instructions. The samples found positive for *M. genitalium* were further analyzed for MRMs by the ResistancePlus^®^ MG assay (SpeeDx Pty. Ltd., Eveleigh, Australia). The latter uses propriety PCR technology to detect *M. genitalium* via the presence of the *MgPa* gene and to detect any of the known mutations in the 23S rRNA gene that are associated with macrolide-resistance (A2058G, A2059G, A2058C, A2059C and A2058T; *E. coli* numbering) [20]. Sanger sequencing of the 23S rRNA and *parC* genes was performed on *M. genitalium*-positive samples for the MRMs’ confirmation and determination of quinolone resistance-associated mutations (QRAMs), respectively [21,22]. In case of treatment failure, post-treatment *M. genitalium*-positive samples were examined for the presence of spontaneously emerged MRMs or QRAMs that are selected during azithromycin or moxifloxacin treatment, respectively.

### 2.3. Treatment Outcome

The treatment outcome was assessed by the treatment failure rate and the mean time to achieve a microbiological cure. The treatment failure rate was calculated as follows: numerator = number of participants with treatment failure at follow-up (defined as *M. genitalium*-positive at TOC, with or without persistent symptoms and with no reinfection risk); denominator = all those treated according to the macrolide-resistance assay. For both the denominator and numerator, only those who were followed-up were included. Time to achieve a microbiological cure was defined as the time (in days) to obtain an *M. genitalium*-negative test from the first positive result. The mean time to achieve a microbiological cure was estimated with 95% CI.

To compare treatment outcomes before and after the implementation of *M. genitalium* diagnostics to macrolide-resistance detection, the treatment failure rate and the mean time to achieve a microbiological cure were compared with data from our previous *M. genitalium* therapy outcome study, in which no macrolide-resistance assay was utilized (i.e., pre-RGT patient group) [19].

### 2.4. Statistical Analyses

First, the treatment failure rate—as the proportion—and the mean time to achieve a microbiological cure with 95% confidence intervals (CI) were calculated, then Fisher’s exact test was used to compare treatment failure rates and the Mann–Whitney U-test was used to compare mean times (days) to achieve a microbiological cure. In the statistical analysis, *p* < 0.05 was considered significant.

### 2.5. Ethical Approval and Informed Consent

The study was conducted following the Declaration of Helsinki and was reviewed and approved by the institutional review board of the National Center of Infectious and Parasitic Diseases (IRB), 00006384. Written informed consent was obtained from all patients for personal data collection and microbiological sample testing.

## 3. Results

### 3.1. Selection of Cases, Treatment Outcomes and Demographic Characteristics

Twenty-two patients were diagnosed with *M. genitalium* during the study period (Figure 1). Four patients were ineligible, as they did not receive first-line therapy according to the results of the macrolide-resistance assay. The causes for not following the macrolide RGT were syndromic management with doxycycline before establishing the diagnosis (*n* = 3) and receiving a single-dose of azithromycin with ceftriaxone for coinfection with gonorrhea (*n* = 1). Of the eighteen enrolled participants, in nine (50%) of the cases, no MRMs were detected, and the corresponding patients received azithromycin as their first-line treatment. The remaining nine cases were found to have MRMs and were treated with moxifloxacin accordingly. Of all of the enrolled participants, seventeen (94.5%) completed all aspects of the study and one participant with detected MRM who was treated with moxifloxacin did not provide TOC samples (lost to follow-up, *n* = 1). Overall, there were no treatment failures in those patients with MRMs-negative infections who received azithromycin. Among the cases with detected MRMs, at TOC, one case of treatment failure was observed. The case involved a patient with a persisting *M. genitalium* infection with no reinfection risk after moxifloxacin therapy. The patient was then successfully treated according to the European *M. genitalium* guideline recommendations (pristinamycin 1 g four times daily for 10 days) [5].

The median age of the patients with *M. genitalium* infection was 29 (age range 18–47); fourteen (82.4%) were men and three (17.6%) were women. Twelve (85.7%) *M. genitalium*-positive males had symptomatic urethritis and 2 (14.3%) were asymptomatic contacts with rectal infections. Two (66.7%) of *M. genitalium*-positive females presented with cervicitis and mucopurulent discharge. The remaining one female was an asymptomatic contact. The comparator group (pre-RGT patient group) consisted of cases with similar sample size and patient demographic characteristics to our previous *M. genitalium* therapy outcome study with no macrolide-resistance detection (Table 1) [19].

### 3.2. Macrolide- and Quinolone-Resistance Mutations (MRMs and QRAMs)

In 2022, macrolide-resistance-associated 23S rRNA gene mutations were detected in 47% (8/17) of the cases by the ResistancePlus^®^ MG assay. These results were then confirmed by 23S rRNA gene sequencing, with further specification of the particular mutation at position A2058 or A2059 (*E. coli* numbering). The mutation A2059G (*n* = 7) was predominating, and the mutations A2058T and A2058G were each found in one case (Figure 2b). The distribution of the wild-type and MRMs was similar in the pre-RGT and RGT groups, i.e., before and after the implementation of the *M. genitalium* diagnostics with macrolide-resistance detection, with an overall macrolide-resistance rate of 48% [19] and 47%, respectively (Figure 2a,b).

During the study period, QRAMs in the *parC* gene were detected in 18% of the cases. The possible resistance-associated ParC amino acid alterations were D87N (*n* = 2) and S83I (*n* = 1). The distribution of the wild-type and QRAMs was similar in the pre-RGT and RGT groups, with an overall spread of the possible quinolone-resistance of 15% [8] and 18%, respectively (Figure 3a,b).

The spread of dual resistance (both MRMs and QRAMs present) was 9.5% and 11.7% among the pre-RGT [8,19] and RGT groups, respectively (Figure 4).

No macrolide- or quinolone-resistance was selected during azithromycin or moxifloxacin treatment, as no further MRMs or QRAMs emerged in the post-treatment positive samples (*n* = 1), in comparison to the pre-treatment sample. Furthermore, the observed treatment failure case involved an infection with the same MRM and QRAM in the pre- and post-treatment samples (i.e., A2058T and S83I).

### 3.3. Treatment Failure Rate and Mean Time to Cure

Seventeen of the eighteen patients given RGT returned for TOC (94.4%), comprising fourteen males and three females. Among them, the overall treatment failure rate was 1/17 (5.9%). This was significantly lower than the treatment failure rate (47.6%) observed in patients treated pre-RGT (*p* = 0.002) [19]. For patients given RGT who returned for TOC (*n* = 17), the mean time to achieve a microbiological cure was 29.4 days (CI 24.5–34.3), compared to 45.2 days (CI 36.5–53.7) in the pre-RGT group (*p* = 0.001).

## 4. Discussion

Implementation of *M. genitalium* diagnostics with macrolide-resistance detection in Bulgaria improved patient treatment outcomes in a population where almost half of the cases are macrolide-resistant. This was achieved by selecting the first-line antimicrobial according to a macrolide-resistance assay, which was performed as an addition to the routine molecular diagnostics. The overall treatment failure rate observed in this study was 5.9%, which was significantly lower than the treatment failure rate (47.6%) in the Bulgarian patients group before the implementation of the resistance assay (*p* = 0.002). The mean time to achieve a microbiological cure was 29.4 days (CI 24.5–34.3) compared to 45.2 days (CI 36.5–53.7) (*p* = 0.001) [19]. These results are consistent with other studies that have clinically demonstrated an improvement in the patient cure rate and a reduction in the time to achieve a cure for *M. genitalium* patient management. In Australia, the implementation of *M. genitalium* diagnostics with macrolide-resistance detection dramatically improved the cure rate to 93% in 2018, compared to 2013 (~40%) [15]. The preliminary rate of successful eradication from Germany (93.3%) was favorable for the continuation of the diagnostic strategy of including macrolide-resistance detection [18]. In the United Kingdom, the treatment failure rate was significantly reduced (3%) compared to before the implementation of *M. genitalium* diagnostics with macrolide-resistance detection (27%) (*p* = 0.008) [17]. Furthermore, there was a trend of a shorter time to obtain a negative TOC in male urethritis (55.1 [95% 43.7–66.4] vs. 85.1 [95% 64.1–106.0] days, *p* = 0.077) [17].

Among those patients who had MRMs-negative infections and received azithromycin, there were no treatment failures observed and, accordingly, no macrolide-resistance was selected during treatment. Nevertheless, mycoplasmas have a high mutation rate and random MRMs may spontaneously emerge in a population of wild-type *M. genitalium* bacteria during RGT [23]. A meta-analysis by Horner et al. found that an extended azithromycin regimen for *M. genitalium* (500 mg on day 1 followed by 250 mg on days 2–5; 1.5 g total oral dose) may be more effective than a 1 g of azithromycin single oral dose, and that it is less likely to cause selection of macrolide-resistance [24]. The absence of selected MRMs during azithromycin treatment in this study may indicate a lower selection rate using the increased azithromycin dose in Bulgaria (2.5 g total oral dose), compared to the azithromycin 1 g single-dose and the azithromycin extended 1.5 g regimen [25].

Although the prevalence of reported macrolide-resistance varies substantially between regions and countries [10], macrolide-resistance has been rapidly increasing and is now above 50% in many countries around the globe [12]. Consequently, *M. genitalium* diagnostics with macrolide-resistance detection is encouraged in most international guidelines [5,26,27,28,29,30]. In this study, a high rate of macrolide-resistance (47%) in *M. genitalium* infections was reported from Bulgaria in 2022. Similarly high rates were also observed in previous years in Bulgaria [8,19]. These high antimicrobial resistance rates in *M. genitalium* cases have emerged in Bulgaria in the context of no or very limited *M. genitalium* testing and no national *M. genitalium* antimicrobial resistance surveillance. Accordingly, no recommendations for patient management (diagnostics and treatment) exist yet in Bulgaria. In most settings, macrolides, particularly an increased azithromycin dose (1 g day 1, 0.5 g days 2–4), have been preferred as empirical first-line treatment. However, that treatment is not effective in resistant strains, and macrolide-resistance detection is performed only at the National Center of Infectious and Parasitic Diseases. The findings of the present study, including the high rate of macrolide-resistance, clearly emphasize that routine macrolide-resistance detection before starting therapy for *M. genitalium* infections is imperative in Bulgaria.

Among those patients who had MRMs-positive infections and received moxifloxacin, one case of treatment failure was observed. The case involved a persisting *M. genitalium* infection after moxifloxacin therapy with no reinfection risk. Molecular analysis revealed that it was caused by an *M. genitalium* strain with dual resistance (both MRM and QRAM detected) and no macrolide- or quinolone-resistance was selected during treatment (i.e., A2058T and S83I detected in the pre- and post-treatment samples). This finding poses a grave concern because there is no highly effective and accessible third-line treatment for *M. genitalium* infections at present. Accordingly, the European *M. genitalium* guidelines recommend pristinamycin, minocycline or doxycycline, and none of these antimicrobials cure all *M. genitalium* cases (with observed cure rates of 75%, 70% and 40%, respectively) [5]. Furthermore, pristinamycin is expensive and it is not available in many countries worldwide, including Bulgaria, and has to be explicitly imported by clinicians.

The prevalence of QRAMs is increasing worldwide [10], and the reported QRAM rates range from less than 5% in northern Europe up to around 20% in southern Europe [12]. However, scientific evidence indicates that not all QRAMs cause quinolone-resistance in vitro, and the association between mutations and treatment failure is not well-established [12]. Accordingly, the most significant QRAM (i.e., S83I) leads to moxifloxacin failure in only 60% of the treated patients, but the absence of the S83I is highly predictive of a moxifloxacin cure (96.4%; 95% CI, 93.7–98.2) [31]. That suggests that incorporating the detection of quinolone-resistance in *M. genitalium* diagnostics would not be as successful in determining the first-line treatment, but rather in individualizing the TOC [32,33]. Furthermore, a novel therapeutic approach (i.e., resistance-guided sequential therapy) has shown higher cure rates and lower selection of resistance in populations with a high prevalence of macrolide and quinolone-resistance [15,16]. That approach comprises sequential therapy by pre-treating with doxycycline and selecting the second antimicrobial with a macrolide-resistance assay. In this study, a high prevalence of QRAMs (18%) in *M. genitalium* infections was demonstrated in Bulgaria in 2022. In the country, a widespread prevalence of QRAMs was likewise observed in previous years [8]. However, the current study reports the first verified case of moxifloxacin treatment failure. Nevertheless, because of the lack of consistency in the association of QRAMs with the treatment outcomes, the detection of quinolone-resistance in Bulgaria is not indicated outside of scientific research. Resistance-guided sequential therapy appears to be a viable approach among Bulgarian patients to delay further emergence and spread of antimicrobial resistance. Ultimately, novel effective and affordable antimicrobials for the treatment of *M. genitalium* infections are essential.

The main limitation of this study was the small sample size from one sexual health clinic in Sofia, Bulgaria. However, given the high reported rate of treatment failures in Bulgaria [19], the study results provide necessary information about preliminary monitoring of treatment outcomes after the implementation of a macrolide-resistance assay until further data with more samples become available. Another limitation was that patients were recommended treatment according to ResistancePlus^®^ MG assay results, which is a slightly less sensitive method than the gold standard of Sanger sequencing [34,35]. Nevertheless, the confirmatory 23S rRNA sequencing performed in the study showed that all eligible participants were correctly allocated for the appropriate treatment by the ResistancePlus assay. A further concern is the increased cost of the *M. genitalium* diagnostics when incorporating the macrolide-resistance assay. In this regard, a recent study in Australia showed that this diagnostic approach is cost-effective for *M. genitalium* infections, supporting its adoption as a national management strategy [36].

## 5. Conclusions

Implementation of *M. genitalium* diagnostics with macrolide-resistance detection improved treatment outcomes in Bulgaria, with significantly lower treatment failure rates and reduced time to achieve a microbiological cure. In light of limited treatment options and concerns about their decreasing efficacy in response to misuse and overuse, a diagnostic macrolide-resistance assay is critical to direct appropriate first-line treatment, to maintain the efficacy of antimicrobial treatment (antibiotic stewardship) and to minimize the spread of antimicrobial resistance.

## Figures and Tables

**Figure 1 diagnostics-14-02665-f001:**
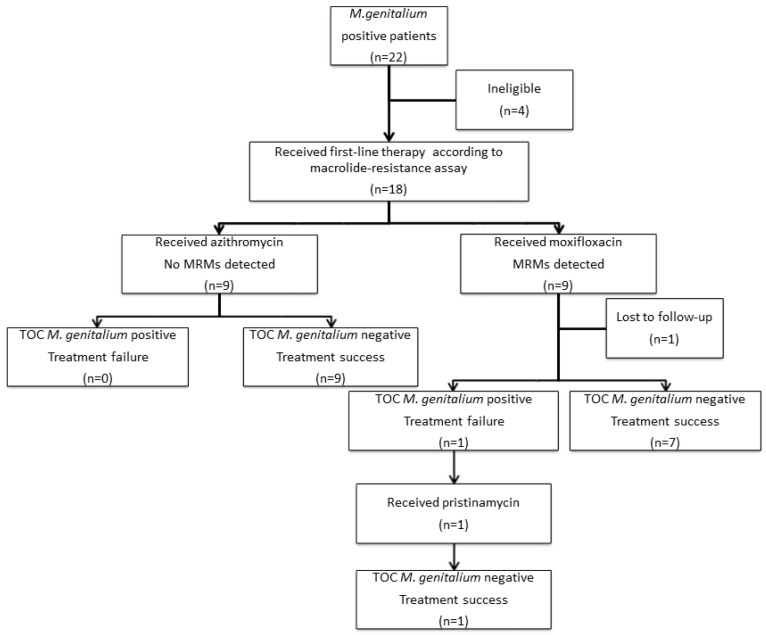
Selection of cases and outcomes of *M. genitalium* therapy guided by diagnostics with macrolide-resistance detection from Bulgaria, 2022.

**Figure 2 diagnostics-14-02665-f002:**
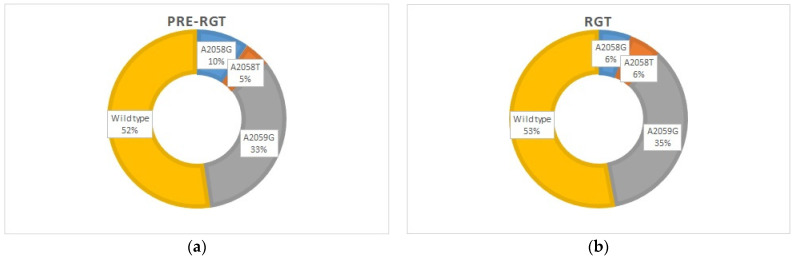
Distribution of wild-type and different MRMs in *M. genitalium*-positive cases: (**a**) before the implementation of the *M. genitalium* diagnostics with macrolide-resistance detection, *n* = 21 (pre-RGT group) [19]; (**b**) after the implementation of the *M. genitalium* diagnostics with macrolide-resistance detection, *n* = 18 (RGT current study group).

**Figure 3 diagnostics-14-02665-f003:**
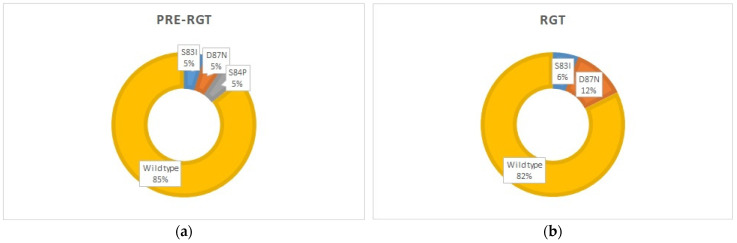
Distribution of wild-type and different QRAMs in *M. genitalium*-positive cases: (**a**) before the implementation of the *M. genitalium* diagnostics with macrolide-resistance detection, *n* = 21 (pre-RGT group) [8]; (**b**) after the implementation of the *M. genitalium* diagnostics with macrolide-resistance detection, *n* = 18 (RGT current study group).

**Figure 4 diagnostics-14-02665-f004:**
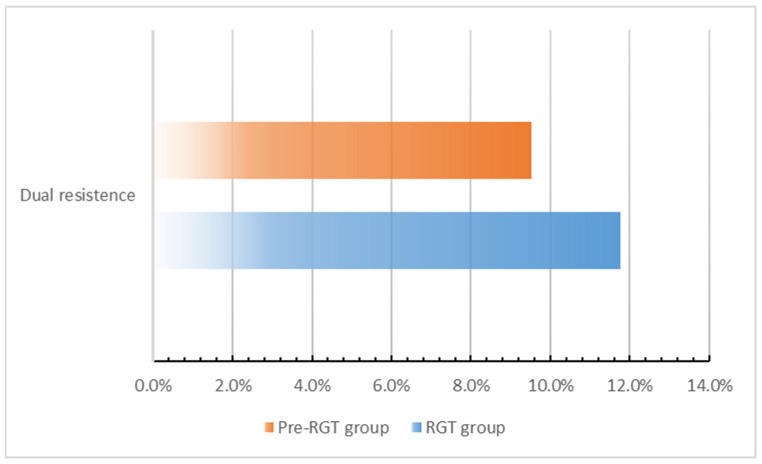
Rates of dual resistance (both MRMs and QRAMs present) before and after the implementation of the *M. genitalium* diagnostics with macrolide-resistance detection (pre-RGT group, *n* = 21 [8,19] and RGT group, *n* = 17).

**Table 1 diagnostics-14-02665-t001:** Characteristics of the patients before (pre-RGT group) and after (RGT current study group) the implementation of *M. genitalium* diagnostics with macrolide-resistance detection.

	Pre-RGT Group [19], *n* (%)		RGT, *n* (%)	
	Male (*n* = 18)	Female (*n* = 3)	Male (*n* = 14)	Female (*n* = 3)
Median age (range)	32 (22–49)	28 (23–33)	29 (18–47)	29 (23–34)
Presentation				
Symptomatic	15 (83.3)	1 (33.3)	12 (85.7)	2 (66.7)
Asymptomatic contact	3 (16.7)	2 (66.7)	2 (14.3)	1 (33.3)
Specimen				
First-void urine	14 (77.8)	0 (0)	7 (50)	0 (0)
Urogenital swab	4 (22.2)	3 (100)	5 (35.7)	3 (100)
Extra-genital swab	0 (0)	0 (0)	2 * (14.3)	0 (0)

* Rectal *M. genitalium* infections.

## Data Availability

The data are available upon reasonable request from the author for correspondence (I.P.).

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
