# Peer review of "Implementation of Mycoplasma genitalium Diagnostics with Macrolide-Resistance Detection Improves Patient Treatment Outcomes in Bulgaria"

_diagnostics, 2024, doi:10.3390/diagnostics14232665_

Round 1

Reviewer 1 Report

Comments and Suggestions for Authors

Dear authors,

The manuscript addresses a very important and actual topic. The results of this study have high applicability for general clinical practice.

However, I have a few suggestions.

1.     Please consider providing more precise data on the prevalence of mycoplasma resistance to antibiotics and treatment failures.

2.     A more thorough discussion on the antimicrobial classes to which mycoplasma have developed resistance in the last decades is advisable.

3.     There is no mention regarding the recommendation for testing and treatment of the patients’ contacts. I recommend expanding the discussion on this issue.

4.     Additionally, the potential complications of M. genitalium infection should be discussed in more detail.

Best regards!

Author Response

Comments 1: Please consider providing more precise data on the prevalence of mycoplasma resistance to antibiotics and treatment failures.

Response 1: Thank you for pointing that out, we agree and have added a couple of sentences that provide more data on the prevalence of antimicrobial resistance in M. genitalium at line 55.

Comments 2: A more thorough discussion on the antimicrobial classes to which mycoplasma have developed resistance in the last decades is advisable.

Response 2: Thank you for pointing that out, we agree and have added a couple of sentences that explain that macrolides and quinolones are the two antimicrobial classes to which M. genitalium has developed resistance in the past years at line 55.

Comments 3: There is no mention regarding the recommendation for testing and treatment of the patients’ contacts. I recommend expanding the discussion on this issue.

Response 3: Thank you for your comment. We agree and have added the European guideline’s recommendations at line 51.

Comment 4: Additionally, the potential complications of M. genitalium infection should be discussed in more detail.

Response 4: We agree with this comment and added more information about the potential complications of the infection at the beginning of the introduction at line 34.

Reviewer 2 Report

Comments and Suggestions for Authors

The authors have evaluated two treatment strategies with pre and post-detection of quinolone and macrolide resistance in M.genitaliuim in patients with infection. It is obvious that detecting the genes prior to treatment would help to give alternate therapy with better outcomes. This has been shown in the study.

limitation of the study, has been mentioned by the authors, is a small sample size

Some of the specific queries are made by comment boxes in the manuscript uploaded.

Author Response

Comments 1: Line 102, “In case of treatment failure, post-treatment M. genitalium-positive samples were examined for the presence of MRMs or QRAMs that are selected during azithromycin or moxifloxacin treatment, respectively”: If these isolates were negative for MRMs or QRAMs in initially (prior to initiation of treatment), mechanism of acquiring resistance during treatment may be discussed.

Response 1: Thank you for your comment. We agree that it needs to be clarified and have incorporated corrections at line 116.

Comments 2: Line 160, Table 1, regarding infections from extragenital swabs: Site?

Response 2: We agree that the site of the extragenital infections should be pointed out and have incorporated a remark at the end of Table 1 that the cases are rectal infections. Furthermore, we made clarifications in the text that the detected rectal infections are asymptomatic contacts at line 167.

Comments 3: Line 226, “mycoplasmas have a high mutation rate and random MRMs may spontaneously emerge in a population of wild-type M. genitalium bacteria during the treatment of the infection”: Mention how would RGT help in these cases.

Response 3: Thank you for pointing this out. We don’t believe RGT per se could help in these cases but rather the optimal dosage regimens (i.e. increased azithromycin dose: 2.5 g total oral dose) is better suited for the prevention of the selection of spontaneously emerged MRMs during azithromycin treatment. We made a clarifying correction to line 241.

Comments 4: Line 249, “clearly emphasize that routine macrolide-resistance detection before starting therapy for M. genitalium infections is imperative also in Bulgaria”: Would the guidelines for increased dosage of Macrolides be an alternate to routine Macrolide resistance detection, in places where it is not available?

Response 4: Thank you for your question. The increased azithromycin dosing regimen cannot be an alternate for macrolide resistance detection because macrolide resistant M. genitalium cannot be eradicated by any dose of azithromycin. We made a clarifying correction to line 259.